# Deep Proteomic Analysis on Biobanked Paraffine-Archived Melanoma with Prognostic/Predictive Biomarker Read-Out

**DOI:** 10.3390/cancers13236105

**Published:** 2021-12-03

**Authors:** Leticia Szadai, Erika Velasquez, Beáta Szeitz, Natália Pinto de Almeida, Gilberto Domont, Lazaro Hiram Betancourt, Jeovanis Gil, Matilda Marko-Varga, Henriett Oskolas, Ágnes Judit Jánosi, Maria del Carmen Boyano-Adánez, Lajos Kemény, Bo Baldetorp, Johan Malm, Peter Horvatovich, A. Marcell Szász, István Balázs Németh, György Marko-Varga

**Affiliations:** 1Department of Dermatology and Allergology, University of Szeged, 6720 Szeged, Hungary; janosi.agi97@gmail.com (Á.J.J.); kemeny.lajos@med.u-szeged.hu (L.K.); nemeth.istvan.balazs@med.u-szeged.hu (I.B.N.); 2Section for Clinical Chemistry, Department of Translational Medicine, Lund University, Skåne University Hospital Malmö, 205 02 Malmö, Sweden; erika.velasquez@med.lu.se (E.V.); johan.malm@med.lu.se (J.M.); 3Department of Internal Medicine and Oncology, Semmelweis University, 1083 Budapest, Hungary; szeitz.beata@phd.semmelweis.hu (B.S.); szasz.attila_marcell@med.semmelweis-univ.hu (A.M.S.); 4Clinical Protein Science & Imaging, Biomedical Centre, Department of Biomedical Engineering, Lund University, BMC D13, 221 84 Lund, Sweden; nataliapalmeida@pos.iq.ufrj.br (N.P.d.A.); matilda.marko_varga@bme.lth.se (M.M.-V.); gyorgy.marko-varga@bme.lth.se (G.M.-V.); 5Chemistry Institute Federal, University of Rio de Janeiro, Rio de Janiero 21941-901, Brazil; gilberto@iq.ufrj.br; 6Division of Oncology, Department of Clinical Sciences Lund, Lund University, 221 85 Lund, Sweden; lazaro.betancourt@med.lu.se (L.H.B.); jeovanis.gil_valdes@med.lu.se (J.G.); henriett.kovacs-oskolas@med.lu.se (H.O.); bo.baldetrop@med.lu.se (B.B.); 7Department of Systems Biology, Faculty of Medicine and Health Sciences, University of Alcala de Henares, 28801 Alcalá de Henares, Madrid, Spain; carmen.boyano@uah.es; 8HCEMM-USZ Skin Research Group, University of Szeged, 6720 Szeged, Hungary; 9Department of Analytical Biochemistry, Faculty of Science and Engineering, University of Groningen, 9712 CP Groningen, The Netherlands; p.l.horvatovich@rug.nl; 10Department of Bioinformatics, Semmelweis University, 1094 Budapest, Hungary; 11Chemical Genomics Global Research Lab, Department of Biotechnology, College of Life Science and Biotechnology, Yonsei University, Seoul 03722, Korea; 12Department of Surgery, Tokyo Medical University, Tokyo 160-8402, Japan

**Keywords:** metastatic melanoma, immunotherapy and targeted therapy responder, prognostic and predictive biomarkers, protein expression pattern in long and short survival, proteomics

## Abstract

**Simple Summary:**

Malignant melanoma is one of the most aggressive cancer types among the solid tumors; therefore, more clinically applicable protein biomarkers predicting survival and therapy response have mandatory importance, impacting patient treatment. The aim of the study was to discover new proteins in biobanked FFPE samples that relate to progression-free survival and response to targeted- and immuno-therapies in patients with melanoma. Protein expressions were detected and quantified by high-resolution mass spectrometry and were integrated with the clinical data and in-depth histopathology characterization. Sample groups with distinct protein expression profiles were connected to longer and shorter survival as well as other clinicopathologic features. In addition, key regulating proteins were assigned, as predictive of progression-free survival in immuno- and/or targeted therapy. Some of the proteins exhibited functionally important correlations to progression and therapy response, which ultimately contributes to a better understanding of melanoma pathology.

**Abstract:**

The discovery of novel protein biomarkers in melanoma is crucial. Our introduction of formalin-fixed paraffin-embedded (FFPE) tumor protocol provides new opportunities to understand the progression of melanoma and open the possibility to screen thousands of FFPE samples deposited in tumor biobanks and available at hospital pathology departments. In our retrospective biobank pilot study, 90 FFPE samples from 77 patients were processed. Protein quantitation was performed by high-resolution mass spectrometry and validated by histopathologic analysis. The global protein expression formed six sample clusters. Proteins such as TRAF6 and ARMC10 were upregulated in clusters with enrichment for shorter survival, and proteins such as AIFI1 were upregulated in clusters with enrichment for longer survival. The cohort’s heterogeneity was addressed by comparing primary and metastasis samples, as well comparing clinical stages. Within immunotherapy and targeted therapy subgroups, the upregulation of the VEGFA-VEGFR2 pathway, RNA splicing, increased activity of immune cells, extracellular matrix, and metabolic pathways were positively associated with patient outcome. To summarize, we were able to (i) link global protein expression profiles to survival, and they proved to be an independent prognostic indicator, as well as (ii) identify proteins that are potential predictors of a patient’s response to immunotherapy and targeted therapy, suggesting new opportunities for precision medicine developments.

## 1. Introduction

Malignant melanoma is the most metastatic human cancer among all tumor types [1,2] and was responsible for more than 50,000 deaths worldwide in 2020 [3].

The incidence of melanoma is anticipated to increase 3% annually at least until 2022 in Norway, Sweden, the UK and the US [4].

While the primary prevention and patient education (e.g., patient factors: knowledge of the disease, educational level, patient–doctor relationship) are better than they were decades ago, the unpredictable behavior of the disease and the maintenance of a proper therapy regimen is still a struggle of the healthcare systems.

Nowadays, the first- and second-line therapy of disseminated melanoma is based on the kinase and immune checkpoint inhibitor approaches. The kinase inhibitor therapies, such as BRAF inhibitors, combined with MEK inhibitors, achieve a dramatic tumor response rate in a short period with the prolongation of the progression-free survival (PFS) and the overall survival (OS). The immune checkpoint inhibitors, such as anti-PD-1, PD-L1 and CTLA4 agents, have a prolonged effect both on the melanoma niche and on the innocent immune cells [5]. Due to the molecular mechanisms of therapy resistance and toxicity, the relapse of the disease is a common phenomenon during the treatment. Even though the effect of targeted and immunotherapy extend the objective response rate of metastatic melanoma patients to approximately 50–70% [4,6], there are no proper predictive biomarkers or tools with high sensitivity and specificity values to predict ad hoc therapy response and distinguish among biological therapies.

Currently, at the genetic level, the mutations of driver oncogenic genes such as BRAF, NRAS, KIT, PTEN, TP53 and NF1 are considered as the mainstay of melanoma development. Interestingly, a previous study revealed that the mutated proteins of the aforementioned genes have an effect on survival [7] and may have an impact on further therapy response.

Moving forward, multiple studies demonstrated the molecular mechanisms behind the new therapies [8,9,10]. In a recent trial, the melanoma patients treated with anti-PD-L1 therapy showed responsiveness with the overexpression of IFN-gamma and IRE genes [1,11]. 

Furthermore, according to the Genomic Classification of Cutaneous Melanoma [12], PTEN mutations, PD-1, PD-L1 and MITF genes were observed in significantly higher copy numbers in BRAF-subtype melanomas defined with the presence of hot-spot mutations; thus, these genetic alterations and the aforementioned immune mechanisms explicate the importance of applied immunotherapies or the need for predictive markers regarding therapy choice and survival. 

This retrospective pilot study is based on a previous analysis that could establish a high-throughput proteomic and phosphoproteomic dataset on formalin-fixed, paraffin-embedded (FFPE) tumor samples compared to analysis of fresh frozen tumor (FFT) samples for the first time. A total of 7629 proteins and 12,659 phosphopeptides were identified, including the phospho-site quantification of the BRAF-MEK-ERK pathway [13].

In this study, we have conducted a comprehensive proteomic analysis on the FFPE melanoma samples to represent the potential molecular changes between samples, address the intratumoral heterogeneity on histopathology validation and detect proteins predictive of therapy outcome and survival. In the pilot study, 90 samples from 77 patients were involved and analyzed for both histopathologic features and for global proteomic analysis (Figure 1). We detected six groups of samples based on unsupervised hierarchical protein clustering and correlated the clusters with clinical and histopathological parameters. We also examined proteomic differences between primary and metastasis samples and identified proteins differentially expressed between the clinical stages. Additionally, protein expressions were evaluated based on whether they represent potential predictive markers for immuno/targeted therapy response. Our study aimed to address the following question: can proteomics provide us with new potential biomarkers or dysregulated pathways that are predictive of therapy response? 

## 2. Materials and Methods

### 2.1. Patient Cohort 

The melanoma samples were collected from the Department of Dermatology and Immunology of the University of Szeged. This study was conducted according to the guidelines of the Declarations of Helsinki, and approved by the Hungarian Ministry of Human Resources, Deputy State Secretary for National Chief Medical Officer, Department of Health Administration. The protocol code is MEL-PROTEO-001, the approval number is 4463-6/2018/EÜIG and the date of approval is 12 March 2018.

In the study, 52 patients with primary and 25 patients with metastatic melanoma were enrolled, 53 primary and 37 metastatic melanoma samples were analyzed, and 8 paired primary-metastasis correlations were compared.

The patients were selected from 2005 to 2020 whose primary melanoma or melanoma metastasis is archived in paraffin-embedded tissue blocks. All primary tumors resulted in loco-regional and/or disseminated disease. The histopathological slides were made from formalin-fixed, paraffin-embedded (FFPE) blocks.

A total of 90 samples were collected with the clinical information, including gender (39 males and 38 females), age at primary tumor (mean = 64.33 yrs, SD ± 10.9, *n* = 77 (in 3 cases the age at primary was not available), age at metastasis (mean = 65.93 yrs, SD ± 10.93, *n* = 77), age at collection date (mean = 64.98 yrs, SD ± 10.81, *n* = 77), localization of primary tumor (trunk = 17, lower limbs = 13, upper limbs = 8, head and neck region = 14, acral region = 1) and metastases, long-term follow up data of the patients: disease-free survival (DFS) (mean = 17 months, SD ± 28, *n* = 77), progression-free survival (PFS) (mean = 42 months, SD ± 39.27, *n* = 77), overall survival (OS) (mean = 51 months, SD ± 45.3, *n* = 77) (in 3 cases the DFS, PFS and OS were not available). Histological subtypes (SSM—superficial spreading melanoma, NM—nodular melanoma, ALM—acrolentiginous melanoma, LMM—lentigo maligna melanoma etc.), pathological TNM staging (according to the 8th edition American Joint Committee on Cancer (AJCC8) cancer staging system), histological parameters of the primary tumor (Clark level, Breslow level, presence of regression and ulceration), BRAF status (WT, BRAFV600E, BRAFV600K and D5879). Twenty-two patients received immunotherapy (CTL4-inhibitors—ipilimumab, PD-1 inhibitors—pembrolizumab and nivolumab, PD-L1 inhibitor—atezolizumab); 15 patients received targeted therapy (BRAFi (dabrafenib and vemurafenib) and MEKi (trametinib and cobimetinib); 59 patients received other therapies (irradiation, chemotherapy, IFN therapy and electrochemotherapy); and 18 patients did not receive any treatments. Three patients received both immunotherapy and targeted therapy. OS was calculated from the date of diagnosis of primary melanoma to the date of last follow-up (long survival group, live status marked with 1, *n* = 34) or death (short survival group, live status marked with 0, *n* = 43), PFS was calculated from the date of diagnosis of primary melanoma to the date of progression, DFS was calculated from the date of diagnosis of primary melanoma to the date of first metastasis. The clinicopathologic data of the samples were collected in an Excel file for statistical analysis. 

#### Survival Analysis of Patient Cohort

Kaplan–Meier (KM) survival analyses were conducted with the disease-free survival (DFS), progression-free survival (PFS) and overall survival (OS) (measured with months) using different statistical approaches (KM including log-rank, Breslow and Tarone-Ware tests). Kaplan–Meier survival analyses and figures showing *p*-values, quartile values, mean values and 95% confidence intervals were produced by IBM SPSS statistics package (26.0 version) software. Alpha was set to 0.05 and *p*-values less than 0.05 were considered significant.

### 2.2. FFPE Sample Preparation and Histology Validation

A total of 90 primary and metastasis samples were collected retrospectively and obtained by excision or biopsy of melanoma or metastatic organs and archived by formalin-fixed and paraffin-embedded (FFPE) routine methodology. Stepwise sectioning of the FFPE tissues was conducted by a conventional microtome. The setting is adjusted to the slice thickness at 10 µm. FFPE tissue sections were placed on glass slides and stained with hematoxylin and eosin. After staining, the slices were placed in an automated slide scanner system (3D Histech Ltd., Budapest, Hungary). The slides were evaluated for the tumor and stroma ratio. 

#### FFPE Sample Processing

The FFPE tumor slides were incubated for 10 min at 97 °C, 500 rpm in 1 mL of EnVision Agilent solution (Agilent, CA, USA) (dilution 1:50). After incubation, the samples were centrifuged (14,000× *g*, 3 min, 4 °C), discarding the supernatants. For a complete deparaffinization, this step was repeated four times until the supernatant was cleared out. Samples were resuspended in 500 μL of protein extraction buffer (25 mM DTT, 10% (*w*/*v*) SDS in 100 mM TEAB pH 8.0) incubating for 1 h at 99 °C in constant agitation (500 rpm). Next, tissues were sonicated (40 cycles, 15 s on/off) in the Bioruptor (Diagenode, Denville, NJ, USA) followed by centrifugation at 20,000× *g* for 20 min at 18 °C. Supernatants were stored at −80 °C until further use, separating a sample aliquot for protein determination (660 nm Protein Assay/Ionic Detergent Compatibility Reagent–Thermo Fisher, Rockford, IL, USA).

### 2.3. Protein Digestion 

Before protein digestion, a spike-in of Lysozyme C protein from Gallus (Sigma-Aldrich, St. Louis, Missouri, USA) was done in each sample for batch normalization (100:1 ratio, Sample/Lysozyme C). Protein alkylation was performed with iodoacetamide 50 mM in dark condition for 30 min at room temperature. S-Trap™ 96-well plate was used for sample digestion (c). First, an incubation with LysC (enzyme: substrate, 1:50) was added to the samples for 2 h, at 37 °C, followed by trypsin (enzyme: substrate, 1:50) incubation overnight at 37 °C. The samples were acidified with 100% formic acid (FA) (~10% final concentration) to stop the digestion process. Peptides were dried in a speed-vac (Thermo Fisher Scientific) and resuspended in a solution of 0.1% trifluoroacetic acid (TFA)/2% acetonitrile (ACN). The quantitative colorimetric peptide assay (Pierce™—Thermo Ficher, Rockford, IL, USA) was used for the determination of peptide concentration. Peptide samples were spike-in with an indexed retention time (iRT) standard (Biognosys, Schlieren, Switzerland) for the control quality of batch normalization. 

### 2.4. LC-MS/MS Analysis 

Two micrograms of the peptides was injected into the UltiMate 3000 RSLCnano system (Dionex, Sunnyvale, CA, USA) using a trap-column (Acclaim^®^ PepMapTM 100, 75 μm × 2 cm, nanoViper 2Pk, Thermo scientific) and an analytical column (PepMap RSLC C18, 2 μm, 100 Å, 75 μm × 25 cm Thermo Scientific) online to Q-Exactive HF-X mass spectrometer instrument (Thermo Scientific). Chromatographic separation of peptides was performed in a linear gradient of 160 min in a flow rate of 300 nL min^−1^, using 0.1% FA (buffer A) and 0.1% FA/80% ACN (buffer B) solution. The solvent gradient was: 0–3 min 2% of B, 3–115 min 25% of B, 115–125 min 32% B, 125–132 min 45% B, 132–140 90% B 140–145 90% B. 

The spectra were acquired in a data-dependent acquisition mode. Spray voltage was set at 1.85 kV. The full scan resolution was 120,000, considering a dynamic exclusion time of 30 s. The AGC and injection time (IT) were 3 × 10^6^ and 50 ms, respectively. The 20 most intense ions were selected for fragmentation using a higher-energy collisional dissociation. The MS2 resolution was 15,000, using an AGC of 1 × 10^5^, 50 ms of IT, and normalized collisional energy of 28 EV.

### 2.5. Database Searching

Data were searched against the UniProt human database (2020/05/26) and two spectral libraries such as the Proteome tools HCD 28 PD and NIST Human Orbitrap HCD using the Proteome Discoverer 2.4 software (Thermo Scientific). Two missing cleavages for trypsin digestion were allowed. The precursor and fragment mass tolerance were set to 10 ppm and 0.02 Da, respectively. Briefly, the pipeline includes two nodes using the spectrum confidence filter tool. The first node, the dynamics modification at the peptide level, includes the methionine oxidation and lysine methylation of peptides. The acetylation, met-loss, and met-loss + acetyl of the protein N-terminal were also set as dynamic modifications at the protein level. Finally, the carbamidomethylation of cysteine was included as a static modification. The second search node considers the methionine oxidation and lysine methylation of peptides and the protein N-terminal acetylation as dynamic modifications. Static modifications such as cysteine carbamidomethylation were also considered. In addition, the Minora Feature Detector and the Feature Mapper nodes were used as a tool for the data search workflow.

### 2.6. Data Normalization and Batch Effect Correction

No imputation was performed on either a peptide or a protein level during data processing. The raw protein intensities were log_2_-transformed and median-normalized (by centering around the global median, which included all non-zero values of the whole dataset). The spike-in protein Lysozyme C, which was added in equal amounts during sample preparation and therefore expected to have constant expression across the samples, exhibited a continuous drift in its intensity according to the injection order, and this batch effect was visible also on the PCA plot of the samples (Appendix A). To correct for this technical factor, a continuous batch correction method implemented in the proBatch R package (v. 1.6.0) [14] was utilized. Firstly, discrete batches were defined based on the injection order: 1–24, 25–49, 50–74, 75–90, and then a non-linear trend was fitted to the Lysozyme C expression values in each batch (Appendix A). The locally estimated scatterplot smoothing (LOESS) span was set to 0.9, leading to a smoother fitted curve, which is less prone to overfitting. This trend, estimated based on the spike-in protein, was then subsequently subtracted from all protein intensity values. The efficiency of the batch correction was checked on the iRT peptides’ intensity, which was added to the samples before the MS measurement in equal amounts. The iRT peptide levels were found to be very similar after batch correction in the samples; moreover, the PCA plot of the samples now showed no separation based on injection order (Appendix A). Moreover, a separation between tumor samples retrieved from cutaneous (Cut) and lymph node (LN) origins arose after batch correction (Appendix A), showing sample grouping according to tissue type. Therefore, the normalized and batch-corrected expression values were used for subsequent analyses. Intensity distributions of the samples before and after median-normalization and after batch effect correction are visualized in Appendix A.

All data post-processing steps and subsequent statistical tests were performed in R v. 4.0.4 using RStudio v. 1.4.1106. Visualizations were made using ggplot2 v.3.3.3 [15], ggbiplot v.0.55 [16], cowplot v.1.1.1 [17], gridExtra v.2.3 [18] and ComplexHeatmap v.2.6.2 [19]. 

### 2.7. Proteomics Data Analysis

Unsupervised clustering of the samples was performed using proteins that have intensity values in at least 50% of the samples. The hierarchical clustering was done on the Z-score normalized protein expression table using Euclidean distance and complete linkage. The Euclidean distance was selected as it gives equal weight to all proteins, and the best results were achieved using complete linkage. Detection of individual clusters was done via the dynamic tree cutting method, a method implemented in the dynamicTreeCut R package (v. 1.63-1), with the following settings: minClusterSize = 0, method = hybrid, deepSplit = 4. The dynamic tree cutting [20] over constant-height tree cutting was selected as it can identify nested clusters (see Appendix A), and it automatically detects the optimal number of clusters. Overrepresentation of any clinical or histopathological characteristics in the sample clusters was determined using one-sided Fisher’s exact test. Alpha was set to 0.05 and nominal *p*-values less than 0.05 were considered significant, but clinical/histopathologic traits with nominal *p*-values between 0.05 and 0.01 were also discussed.

For the proteomic characterization of the sample clusters, a one-way Analysis of Variance (ANOVA) test with pairwise Tukey’s honest significant difference (HSD) post-hoc tests were performed to identify proteins showing differential expression between the sample clusters. No prior filter for valid values was applied on the expression table, but statistics was performed only on proteins that were quantified in at least 80% of the minimum 2 sample clusters. Benjamini–Hochberg adjusted *p*-values values (FDR) of ANOVA tests less than 0.05 were considered significant. Unsupervised hierarchical clustering of the top 1000 most differentially expressed proteins was performed using Euclidean distance and complete linkage, followed by dynamic tree cutting with the settings minClustersize = 50, method = hybrid, deepSplit = 0. Pathway enrichment analysis of the resulting protein clusters was conducted using Kyoto Encyclopedia of Genes and Genomes (KEGG) [21] and Reactome [22] databases.

Proteins showing differential expression between primary and metastasis samples were examined via *t*-tests (independent *t*-tests for the whole cohort and paired *t*-tests for the paired primary and metastasis samples). Only proteins with min. 80% valid values in both groups were included in the analysis. For the paired *t*-test, the log2 fold changes (FCs) were calculated by calculating log2FC between the paired samples, and then taking the average of those paired differences. Proteins were ranked based on the log2FC × −log10 nominal *p*-value for the subsequent pre-ranked gene set enrichment analysis (pGSEA). Linear relationship between stage and protein expression was assessed using Analysis of Covariance (ANCOVA), where both stage as a numerical variable (2, 3 and 4) and tumor type as a categorical variable (primary or metastasis) were included in the model. Only proteins with min. 80% valid values in all three stages were considered in the analysis. Regression coefficients for stage and nominal *p*-values were extracted, which were used to rank the proteins (coefficient × −log10 nominal *p*-value) for the pGSEA. The pGSEA was performed via the clusterProfiler R package [23]. The GO biological process, KEGG, Reactome and Wikipathways gene sets were utilized for the analysis. The gene sets were downloaded from the Molecular Signatures Database v7.4 [24,25,26]. For primary vs. metastasis comparisons, the FDR-corrected pGSEA *p*-value < 0.05 was considered significant, whereas for stage comparisons, nominal *p*-value < 0.05.

Proteins that are potential predictors of long or short progression-free survival after the patients received immunotherapy or targeted therapy were detected via survival analysis. For this, multiple Cox regression models were fitted to the progression-free survival data in both the immunotherapy and targeted therapy subgroup. Individual protein expression data were used as a covariate in the model (provided that the protein had min. 80% valid values within the therapy subgroup’s samples), and samples were stratified based on the tumor type (Primary or Metastasis). Alpha was set to 0.05 and nominal *p*-values less than 0.05 were regarded as significant. Proteins where the proportional hazards assumption was not met (checked by the cox.zph function of the survival R package, *p*-value < 0.05) were deleted from the list of significant proteins. Protein networks were drawn via STRING (v.11.5) [27]. Overrepresentation analysis of GO biological processes [28] and KEGG pathways, as well as interaction network construction (confidence cutoff 0.4) using the significant proteins, was performed in Cytoscape (v.3.8.2) [29]. The processes showing FDR < 0.05 were visualized and used for biological interpretation.

### 2.8. Data Availability

The mass spectrometry proteomics data have been deposited to the ProteomeXchange Consortium via the PRIDE [1] partner repository with the dataset identifier PXD028930. Project name is Proteomic Analysis on Paraffine Archived Melanoma. Project accession is PXD028930, project DOI is not applicable, username is reviewer_pxd028930@ebi.ac.uk, and password is TaHGkBGm. 

Proteomic data summary, ANOVA and Tukey’s HSD test results of sample cluster comparisons, Cox regression analysis results for immunotherapy and targeted therapy subgroups, *t*-test results (paired/independent) for primary vs. metastasis comparisons as well as ANCOVA results for the comparison of disease stages are provided in Appendix A. The scripts used for the proteomic data normalization, batch effect correction and statistics are available at https://github.com/bszeitz/MM_pilot (accessed on 21 November 2021). Clinical information of individual patients cannot be provided due to the ethics restrictions. 

### 2.9. Figure Illustration

Some of the figure illustrations were created with BioRender 2021 software.

## 3. Results and Discussion

### 3.1. The Clinical Significance of the Study

#### Clinicopathologic Characteristics of the Patient Cohort

The applied retrospective cohort is composed of 90 melanoma samples from 77 patients with mainly progressed melanoma. The samples consist of 53 primary and 37 metastatic melanoma samples representing both lymphatic and cutaneous metastases and eight pairs of primary-metastasis correlation. The clinicopathologic data of the samples are summarized in Table 1. From the investigated cases, 51% were male individuals and 56% of the participants were not alive at the collection date (short survival group). The melanomas were stratified according to the 8th edition of the American Joint Committee on Cancer (AJCC8) staging system [30] and the 3rd edition of the WHO classification [31]. The metastatic cohort had a predominance of patients who had stage III (40%) at the time of primary tumor sampling (Table 1). Furthermore, data of the BRAF mutation were also collected from the primary and metastatic samples, with 51% coming from the mutated BRAFV600 state. 

The 90 analyzed melanoma samples included 58.9% primary tumors, 26.66% locoregional lymph node metastases and 14.44% cutaneous metastasis samples. The majority of the samples were surgically removed from the trunk (32%) and the dominant histotype of the primary tumors were nodular melanoma (51%). The tumors spread into the deeper layers of the dermis or subcutaneous layer of the skin in 51% of the primary samples (Breslow thickness > 4 mm) (Table 1).

The examined FFPE tumor samples varied from medium-thick (1–2 mm–pT2) to thick (high risk: 2–4 mm–pT3, very high risk: >4 mm–pT4) melanomas with a dominance of the thick/very high-risk cases up to even 22 mm of thickness. The primary melanomas mainly showed a poorly differentiated solid area reaching at least Clark III level, indicating a vertical growth pattern even at superficial spreading or lentigo melanoma cases. The peritumoral niche varied from lymphocyte- and macrophage-rich areas to regressive, fibrotic, pauci-cell microenvironment. Focal necrotic areas as well as intratumoral pigmentation were assessed. In the lymphatic and cutaneous metastases, variable degrees of alteration in necrotic areas, viable tumor counterparts, tumor burden, and regressive signs were observed. In addition, the tumor/stroma ratios were evaluated in each sample. (Figure 2).

### 3.2. Proteomic Analysis Reveals Sample Clustering According to Relevant Histopathological and Clinical Parameters Associated with Survival and Tumor Progression

A total of 90 samples from 77 patients were analyzed based on their clinical- and global proteomic expression data. Mass spectrometry analysis quantified 7881 protein groups across all datasets (Appendix A). The unsupervised hierarchical cluster analysis revealed six major sample clusters and one outlier sample labeled as Cluster 0 (Figure 3A). A total of 4437 protein groups showing variability among these clusters were detected (Figure 3A). Enrichment analyses revealed a correlation of one of each cluster to relevant clinical parameters. Cluster 1 corresponded to better overall survival in the interval of 101 to 205 months (Fisher test *p* < 0.05), Cluster 2 corresponded to samples with the host tissue of cutaneous origin, exclusively (Fisher test *p* < 0.001), and a tendency for poorer disease-free survival ranging from 11 to 30 months (Fisher test *p* = 0.0835) was observed. Cluster 3 showed a borderline significant (Fisher test *p* = 0.0913) enrichment of better progression-free survival values (between 61 and 100 months). On the other hand, samples in Cluster 4 had the host tissue of lymphoid origin (Fisher test *p* < 0.001), as well as enrichment of the AJCC8 St. IIID stage (Fisher test *p* < 0.05) and the Breslow level between 4.1 and 8 mm (Fisher test *p* = 0.0616) was observed. Moreover, the dominant gender was male (Fisher test *p* = 0.0703), and a borderline significant association with poor disease-free survival with less than 10 months (Fisher test *p* = 0.0778) was also noted. In Cluster 5, three out of five samples, all primary tumors, can be categorized into the ALM subtype (Fisher test *p* < 0.05). Moreover, the gender was exclusively female (Fisher test *p* < 0.05). The AJCC8 St. IIA (Fisher test *p* = 0.0647), progression-free survival between 11 and 30 months (Fisher test *p* = 0.0662), and overall survival between 11 and 30 months (Fisher test *p* = 0.0662) were also borderline enriched. Cluster 6 corresponded to a better disease-free survival ranging from 31 to 60 months (Fisher test *p* < 0.05); however, these samples exhibited low overall protein coverage, and can therefore be regarded as outlier samples (Appendix A). Similarly, Cluster 0 was built up from one outlier sample; therefore, associations with clinical and histopathological characteristics were not evaluated (Figure 3A).

To unveil the molecular signature behind the sample clusters, we performed a biological pathway analysis considering the top 1000 most variable proteins (ANOVA FDR < 1.05 × 10^−4^). Figure 3A shows the protein clusters obtained by hierarchical clustering, and Figure 3B shows the main biological functions associated with the sample clusters. One of the main findings was the downregulation of protein cluster 2 in sample cluster 4 and their upregulation in sample clusters 5 and 6. The closer data analysis revealed that most of the proteins are related to the cellular matrix function, such as desmoglein-1, desmocolllin-2, and dystonin, specifically in the desmosome and the hemidesmosome assembly. The decrease in proteins related to desmosome are linked to tumor progression in several types of cancers [32,33]. 

Furthermore, for example, desmoglein-1 was proposed as a new therapy candidate due to its participation in the invadopodia formation, promoting tumor invasion and metastasis development [34]. Divergent results have been found concerning the desmocolllin-2 role in metastasis. Evidence suggests that the reduction in the expression of this protein promotes cellular migration and tumor invasion [35,36,37], while other studies affirm that the upregulation of desmoglein 2 promotes vasculogenic mimicry and metastatic spread in melanoma [38]. On the other hand, the decline of proteins associated with the hemidesmosome components was also related to metastasis progression and poor clinical outcomes [39], and it correlates with our results since we have seen the downregulation of these proteins in sample cluster 4 with poor disease-free survival.

Protein clusters 1, 4, 5, 6, 7 and 10 stand out by the dramatic decrease in the sample cluster 6, the better prognosis group. Alterations in mitochondrial translation and RNA metabolism are the most enriched pathways. Detailed examination of the biological significance of these protein changes displays the alteration of the mitochondrial ribosomal complex (e.g., MRP-S7, MRP-L12, MRP-L15) and RNA regulation (e.g., integrator complex). Furthermore, mitoribosomal proteins (MRP) are reported as key drivers in regulating apoptotic signaling beyond their classical protein translation role [40]. Interestingly, the abnormal protein expression of MRP has been described in several tumors, and currently, they are proposed as novel targets for disease diagnosis and treatment [41]. In the same way, the alteration of the MRP-S7 gene expression is strongly correlated to the metastatic process in the osteosarcoma [42], while MRP-L12 was proposed as a potential predictive biomarker for glioblastoma [43]. In addition, MRP-S15 was suggested as one of the predictor markers of tumor recurrence and treatment failure in breast cancer patients [44]. In parallel, the detriment in the abundance of the important components of transcription machinery previously linked to cancer malignancy, such as the subunits of the integrator complex [45].

On the contrary, protein clusters 3, 8, and 11 are highly increased in sample cluster 6. These protein groups are mainly involved in the RNA metabolism in response to DNA damage, especially in the increase in proteins related to TP53/P53 response (e.g., probable ATP-dependent RNA helicase DDX5, heterogeneous nuclear ribonucleoprotein K, 40S ribosomal protein S3). For example, it was demonstrated that 40S ribosomal protein S3 protects p53 against MDM2 ubiquitination and proteasome degradation [46]. In addition, we noticed the increment of proteins linked to phagocytosis and membrane remodeling, such as merlin, reported as a tumor suppressor reducing the migration and proliferation of metastatic melanoma [47]; this finding is in agreement with the clinical characteristics of sample cluster 6 with better disease-free survival.

Similarly, we also discovered an increase in proteins within the complement system, and coagulation cascade (e.g., complement C3, fibrinogen), which are part of the tumor microenvironment, playing crucial functions in promoting tumorigenesis and a scaffold for the binding of tumor cell growth [48,49,50].

### 3.3. Imbalance of Immune-Related Proteins as Molecular Fingerprint of the Sample Clusters with Varying Survival Outcomes

The bioinformatics analysis of the proteomic profiles captures two different sample clusters with distinct overall survival (i.e., cluster 1 and cluster 5). The comparison between cluster 1 vs. cluster 5 detected the downregulation of 316 protein groups (Figure 4A), mainly linked to the immune system (e.g., CD33, ARGI1, NKAP). In this case, the imbalances of these immune-related proteins were associated with poor clinical outcomes. For instance, the alteration in the abundance of proteins such as CD33, a transmembrane receptor expressed by myeloid lineage, was correlated with poor therapy responsiveness and disease progression [51,52]. On the other hand, the increase in ARGI1 in the tumors usually correlates with metastases and worse clinical prognosis [53]. Moreover, the expression of NKAP is negatively correlated with the levels of Notch target genes, which are essential for T-cell development [54]. Moreover, the NKAP acts as an oncogene and is connected with a poor outcome in cancer patients. It was demonstrated that NKAP knockdown suppresses the proliferation of HCT116 and HT-29 cells, inducing apoptosis [55,56]. On the other hand, 74 protein groups were found upregulated. Similarly, we detected the increase in proteins related to the immune system with anti-tumorigenic effects, such as GTPase IMAP family member 5 (GIMA5), described as a promoter of CD4+ T-cell survival and homeostasis [57,58]. GIMA5 also suppresses the proliferation of lung cancer cell lines [59]. In addition, we observed the increment of proteins such as mitogen-activated protein kinase 2 (MAP4K2), usually expressed in B cells of the germinal center and macrophages, which activates the c-Jun N-terminal kinases [60]. Similarly, we detected the different expression of PIK3CA between the sample clusters. This protein participates in several signal transduction pathways, such as the regulation of the T-cells, and cell signaling, but especially those that impact cell proliferation, survival, and apoptosis in tumors [61,62].

### 3.4. Protein Translation and Immune System Stand out in the Proteome Profile of the Sample Clusters with Different Progression-Free Survival 

The comparison of sample clusters (Figure 4B) with different progression-free survival (cluster 3 vs. cluster 5) shows a decrease within 333 protein groups. The translation pathway was found highly enriched, together with the differential expression of several immune-related proteins. As mentioned above, some of the ribosomal proteins that we identified as differentially expressed between sample clusters in this study also have an alternative function to protein translation, such as MRP-L12 (previously described) as well as RPS12 that triggers metastasis development through the activation of the Akt/mTOR/c-Myc signaling pathway in cervical cancer cell lines [63]. Moreover, we can also highlight the decrease in key immune-regulating proteins, such as l-amino-acid oxidase (OXLA), a critical immune regulator inhibiting the proliferation of the peripheral blood mononuclear cells and CD8+ T-cells. The immune response is thereby suppressed in the tumor microenvironment, contributing to tumor cell invasion [64]. In parallel, we also detected the increase in 108 protein groups, mainly related to the growth signaling cascade such as the SHC-transforming protein 1, previously reported as an important factor in the metastatic progression of breast cancer [65]. Moreover, we detected proteins such as E3 ubiquitin-protein ligase (RNF114) and mitogen-activated protein kinase kinase kinase 7 (MAP3K7) involved in mechanisms related to tumor necrosis factor receptor-associated factor 6 (TRAF6), a well-established protein that plays a critical role in melanoma metastasis [66]. RNF114 regulates the ubiquitination of TRAF6 [67], while TRAF6 mediates the signal transduction of MAP3K7 in the activation of the nuclear factor-kappa B [68].

### 3.5. Proteome Mapping Detects Gradual Proteomic Changes in Sample Clusters Exhibiting Different Disease-Free Survival 

To evaluate the proteins that could potentially impact the disease-free survival of our sample cohort, we contrasted the protein changes across three sample clusters with good (Cluster 6) median (Cluster 2) and inferior (Cluster 4) clinical endpoints. Despite sample heterogeneity, data analysis shows a significant alteration of 90 protein groups, of which 49 and 7 proteins tend to increase and decrease across these sample clusters, respectively. 

Figure 4C displays an example of the main differential expressed proteins between the sample clusters in our analysis. Proteins with a tendency to increase are mainly related to RNA processing, such as ribosomal proteins. Similarly, we also detected a key marker such as TRAF6, as well as relevant proteins for the cell cycle control and apoptosis such as cyclin-dependent kinase 2 (CDK2), crucial for the proliferation of melanoma cells [69,70], and armadillo repeat-containing protein 10 (ARMC10) linked to cell proliferation and the inhibition of p53 [71,72]. A gradual decrease in apoptosis-inducing factor 1 (AIF1), a well-known proapoptotic factor, was found to be aligned with early metastasis [73], in addition to peptidyl-prolyl cis-trans isomerase F, an essential component of the mitochondrial permeability transition pore [74].

### 3.6. Proteomic Differences with Regard to Tumor Type and Clinical Stage

In the metastatic cohort, the primary melanoma and their corresponding metastasis samples from eight patients (paired dataset) were compared based on the differences of their proteomic expression profile (via paired *t*-tests); additionally, differences in primaries and metastases from the whole cohort (full dataset) were examined by independent *t*-tests. Pre-ranked Gene Set Enrichment Analysis (pGSEA) outlined a strong agreement between the two results (Appendix A), indicating that primary vs metastasis differences found in paired samples is globally true for the primary vs metastasis differences in the full dataset as well. The B-cell receptor signaling pathway, endoplasmic reticulum calcium ion homeostasis pathway and the chromatin organization pathway were significantly activated (pGSEA *p*-value < 0.05) in metastatic tumors according to both the paired and full dataset, whereas keratinization pathways, epithelial and epidermal cell differentiation pathways, the epidermis development pathway, collagen fibril organization, the extracellular matrix organization pathway, the myeloid leukocyte mediated immunity pathway and the neutrophil degranulation pathway were strongly activated (pGSEA *p*-value < 0.05) in primary tumors. 

In connection with the clinical stages, we have also identified proteins showing up-and downregulation from AJCC8 stage II to stage IV while accounting for tumor type (primary/metastasis differences). A total of 147 proteins were significantly upregulated, and 152 proteins were downregulated in the higher clinical stage (Appendix A). Pathway analysis through pGSEA showed a concordant upregulation of proteins in the formation of humoral immune response, the regulation of immune effector process, the regulation of humoral immune response, and the regulation of complement activation such as CD46, TRAF6 as well as MCM9 protein in DNA repair mechanisms. The downregulation of the PELO protein in epithelial cell differentiation and the PSMD3 protein in the negative regulation of the canonical WNT signaling pathway (Appendix A) was also detected. It is known that CD46 contributes to complement activation and humoral immune response [75], and the overexpression of this protein correlates to the suppression of IFNγ and TNFα production on γδ T-cells and it can further contribute to the formation of a pro-tumor environment, indicating progression through clinical stages. [76,77,78,79] The elevated expression of TRAF6 protein during the progressed melanoma course is parallel with our previous results where this protein was upregulated in sample clusters with enrichment for short survival. The MCM9 is involved in the DNA repair mechanisms, and it was recently shown that MCM9 protein is highly expressed in metastatic melanomas compared to primary melanomas in the TCGA cohort (TCGA [80]) and is also related to the T stage (histopathologic stage) of the tumor [81].

A downregulated tendency of the PELO protein in the epithelial cell differentiation pathway was seen. This protein has a role in cell cycle control and an unfavorable prognostic marker in breast cancer [82]. The downregulation of PSMD3 protein in the negative regulation of the canonical WNT signaling pathway was also seen from the mild to the severe stages. Interestingly, in other studies, the downregulation of PSMD3 was associated with longer survival in glioblastoma multiforme [83] and in melanoma metastasis [84], which is the opposite tendency to what we have seen in our results in the clinical stages. 

### 3.7. Histopathologic Relations to the Clustered Sample Cohort

Although the examined high/very-high-risk metastatic setting is generally supposed to indicate a worse prognosis, the proteomic assays could differentiate into clusters based on survival and therapy-response variables among these mainly aggressive cases. Interestingly, neither of the standardized histopathologic parameters (pT, regression, ulceration, etc.) has proven significant differences among the clustered groups, which indicates an independent prognostic value of the deep proteomic assay, as shown in Figure 5.

### 3.8. Proteins and Pathways Predicting Therapy Response

#### 3.8.1. Proteins and Pathways Indicating Worse Outcome in the Immunotherapy-Treated Subgroup

Subgroups of patients were selected based on their primary treatment (immunotherapy, targeted therapy). Twenty-four samples from 22 patients were involved in the immunotherapy subgroup and 20 samples from 15 patients were used in the targeted therapy subgroup. The samples involved in the pilot study were naive to any treatment arms, whilst we are able to observe the protein expression profile prior to any given therapy. Aiming to identify proteins that are possible predictors for the length of progression-free survival (i.e., worse or better response), multiple Cox regression models were established for the progression-free survival, using the protein expression data. The model was stratified based on sample type, i.e., primary or metastasis. The analysis resulted in 401 proteins in the immunotherapy subgroup and 260 proteins within the target therapy subgroup, which were found to significantly correlate with survival (Multiple Cox regression *p*-value < 0.05), respectively. The significant proteins are visualized within the heatmap, as shown in Figure 6A,B.

The functional analysis of the proteins (Gene Ontology terms, KEGG pathways, Reactome, Wikipathways) (Appendix A) revealed distinct activated pathways for patients with short/long PFS. For the subgroup with shorter PFS after immunotherapy, the significantly upregulated proteins were related to cellular and metabolic processes (Appendix A), also including the VEGFA-VEGFR2 pathway (Figure 6 (2D)) (KEGG pathway database: FDR < 0.05). The VEGFA and its corresponding pathways activate the endothelial cell growth and vasculogenesis that contributes to the metastatic potential of the tumor [1]. The presence of the VEGFA also suggests a prognostic signature for locoregional lymphatic metastasis [1]. In a study, the PD-1/PD-L1 and VEGFA/VEGFC expressions were compared in lymph node metastases to investigate their respective expressions. The study revealed that despite PD-1 expression being correlated with a higher survival rate, VEGFA was connected to the worse prognosis. However, the immunotherapy linkage was not investigated in this case [85]. Our data indicate that an activation of the VEGFA-VEGFR2 pathway can not only be associated with the aggressive behavior of the tumor, but also with the lack of efficient response to immunotherapy. Currently, there are VEGFA blockers used in anti-cancer therapies. In lung cancer, the application of the anti-angiogenic drugs is a potential choice [86]. The humanized anti-VEGF monoclonal antibody (bevacizumab, Avastin) is the first-line treatment in metastatic colorectal cancer and has already been approved by the FDA [87]. These results raise questions about the role of the VEGFA blockers in the case of melanoma. 

In connection with the VEGFA-VEGFR2 pathway, the nitric oxide synthase 3 (NOS3) was also worth noting, since the NO production of this enzyme promotes the VEGF pathway induced angiogenesis [88], tumor proliferation and progression [89]. In line with the previous findings, the downregulation of NOS3 expression (Cox regression test *p* < 0.05) was found in patients with longer progression-free survival with immunotherapy treatment (Figure 6 (2D)).

A similar tendency could be seen regarding RNA splicing. Proteins involved in RNA splicing mechanisms were upregulated in those patients that received immunotherapy, with a lack of tumor response to treatment (i.e., started to progress after a few months). Among proteins with an important role in the alternative splicing processes, we can highlight the significantly upregulated SNRPB2, SNRNP70 and SNRPA1 proteins [90] (GO Biological Process, FDR < 0.05) (Appendix A) (Figure 6 2D). Studies have confirmed that the abnormal expression of the splicing proteins can lead to inappropriate splicing mechanisms, causing tumor progression [91]. Denga et al. also suggested that these disordered splicing mechanisms could promote the loss of cell surface antigens, which are crucial in melanoma and immune cell fusion that might contribute to immunotherapy resistance [91]. In the clinical field, the anti-spliceosome therapies are currently debated in their application of spliceosome machinery targeting [92]. There are several investigations on small molecules which inhibit the steps of the splicing mechanisms, but the usefulness and clinical safety of these approaches are still under investigation [92,93]. Our results can support the importance of studies that identify anti-spliceosome drugs and can provide insight into the protein mechanisms related to immunotherapy resistance.

#### 3.8.2. Proteins and Pathways Indicating Better Outcome in the Immunotherapy-Treated Subgroup

We also identified proteins that are potential predictors of improved response to immunotherapy (Figure 6 (2C)). In this subgroup of proteins, the stroma-induced signatures and the components of the immune system were significantly overrepresented. In the subgroup with better response to immunotherapy, the significantly upregulated proteins and pathways were enriched in functions such as neutrophil degranulation, complement cascade, B-cell differentiation, neutrophil-mediated immunity, extracellular (EC) organization, ECM–receptor interaction, (for example, ARGN protein), cell adhesion (for instance, ICAM2 protein)*,* integrin cell surface interactions and the PI3K-Akt pathway, the components of which are related to cell adhesion (represented by proteins such as COL4A2 and COL6A2) (KEGG pathways, GO biological processes, FDR < 0.05) (Figure 6 (2C)). In our study, those proteins that were incorporated in the neutrophil degranulation, such as proteinase 3 (PRTN3) [94,95], showed an upregulated pattern aligned with the slow and late development of the progressive behavior of the tumor with a better outcome. In agreement with our results, Babačić et al. revealed that during anti-PD-1-R therapy, increased neutrophil degranulation was detected in the plasma and consequently the neutrophil to lymphocyte ratio was correlated to the longer overall survival in this in-depth plasma proteomics study [96]. Besides the immune response, the function of the EC organization in the tumor microenvironment is not negligible. There are several studies that highlight the coupled functions of ECM and the immune system against melanoma. For instance, in a recent study, Fejza et al. presented that the ECM proteins are found to be associated with the efficacy of the PD-1 treatments [97]. Interestingly, in our study, the EC proteins were upregulated, mostly in cell adhesion components, in the late progressive phase of the tumor development and linked to the longer survival rate as well as the response to immunotherapy. Furthermore, the ECM serves as a basis of the pre-metastatic niche, which is a new and unidentified target regarding the anti-cancer therapies. A recent study summarized those mechanisms that can be crucial in preventing the development of “the metastatic soils” leading to arrest of the tumor invasion [98]. For instance, the prevention of EC vesicles such CXCR2 and CXCR4 inhibitors, vascular stabilization and the immunomodulation can lead to the demolition of the complicated mechanisms in the stroma signature [98]. Further investigation is needed to strengthen the role of both the ECM’s and the microenvironment’s role in tumor management.

#### 3.8.3. Proteins and Pathways Altering the Patient Outcome in the Targeted Therapy Subgroup

The responsiveness to the targeted therapy can arise from abundant mechanisms, such as the upregulation of survival pathways, reactivation of MAPK or the presence of a dedifferentiated/MITFlow cell state [99]. In our functional analysis of proteins that indicate worse prognosis by their overexpression, the metabolic pathways were significantly enriched (KEGG pathways, GO biological processes: FDR < 0.05) (Figure 6 (2E)). Specifically, the MAPK signaling pathway elements (such as PAK4 and MAP2K2 proteins), mTOR pathway members (such as PTEN and DEPTOR proteins), heat shock response proteins and cyclin-dependent kinase 5 were enriched to the same extent in the metabolic pathways and implicated in the resistance mechanisms in the targeted therapy subgroup in our pilot study. The reactivation of the MAPK pathway is reported in up to 60% of melanoma [99,100,101,102], intrinsically linked to our results, in which the proteins concerning the MAPK pathway showed upregulation in the early progressors in the targeted therapy subgroup. Proteins pertaining to the PTEN and mTOR signaling pathways, also implicated worse outcomes during the administration of targeted therapy. These findings correspond to several studies, demonstrating that the activation of mTOR accelerates the proliferation and survival mechanisms of the melanoma cell, thus leading to resistance to BRAF inhibitors [99,103,104]. Genetic alterations of PTEN can be found in 30% of melanomas and also are associated with the BRAF V600E mutations [99,105,106,107]. As a consequence of these genetic changes, the upregulation of the PI3K-PTEN-Akt might promote the resistance mechanisms to targeted therapies in some cases [108]. Intriguingly, in our patient cohort, the PTEN expression pattern was also the strongest in three patients whose tumor progressed in five months alongside targeted therapy. Clinicopathologic similarities were also observed in these three cases. The Breslow level of these primary melanomas were more than 4 mm with histopathologic classification of pT4 (AJCC8). All of these were nodular melanomas with a short progression-free survival. The behavior of these three melanomas can be described as aggressive in parallel with the fast progression rate throughout the therapy. These mechanisms may contribute to the complicated resistance mechanisms in melanoma during treatments and to ascertaining the hidden effect of the metabolic pathways. To verify these results, further investigations including a larger patient cohort are needed.

In the functional analysis of predictive proteins providing better response to targeted therapy, we found significantly up- and downregulated pathways such as mRNA processing, RNA degradation, metabolism of RNA and proteins referring to spliceosomes (KEGG pathways, GO biological processes, FDR < 0.05) (Figure 6 2F). Intriguingly, the RNA splicing mechanisms were also observed in the analysis of the immunotherapy subgroup, but there, activation of these processes was linked to worse prognosis. Our data thus suggests the ambivalent role the RNA mechanisms potentially might have: (i) acting as predictors of better response in patients receiving targeted therapy; (ii) predictors of worse response in patients receiving immunotherapy. However, it is noteworthy that the proteins involved in RNA mechanisms in the targeted therapy subgroup are not identical to the ones within the immunotherapy subgroup. Moreover, we have seen a downregulation of proteins in cell adhesion and ECM organization in early progressed tumors, an observation also made in the immunotherapy subgroups. Thus, our results indicate that the overexpression of the proteins in EC mechanisms can be associated with good prognosis and may predict a better therapy response. These data may raise questions about the real role of the stroma in the melanoma response during the application of certain treatments.

We acknowledge that our metastatic cohort study has some limitations, such as the heterogeneity of the samples due to the different histotypes with varying histopathologic characterization and clinical parameters. Furthermore, tumor content of the samples involved in the therapy subgroups are ranging from a few percent up to 90%, which means that significant mechanisms incorporating therapy response can be related to pathways in stroma cells. The upregulation of EC components in some samples gives us hints about the effect of tumor content differences on the proteomic profiles. On the other hand, our findings can form the basis of further investigations highlighting the role of the tumor–stroma interactions on the progression of the melanoma.

## 4. Conclusions

As melanoma malignum constitutes a highly heterogeneous tumor morphology, diagnosing and high precision accuracy are major challenges.

Beyond the well-known predictive proteins such as mutant BRAF kinase and plasma lactate dehydrogenase (LDH), which are used on a daily basis worldwide, additional protein biomarkers are necessary to unravel resistance mechanisms and predicting the efficacy of the treatments.

The proteomic analysis of 90 FFPE melanoma samples revealed the discovery of predictive protein candidates related to the progression-free survival as well as the therapy responsiveness. The histopathologic analysis, clinical parameters of the patient cohort and the expression pattern of the proteins were outlined. We found that certain mechanisms and protein expressions in melanoma aligned with disease progression. For instance, in sample clusters with enrichment for shorter survival, TRAF6, ARMC10, CDK4, ITGA5, CAMK4 and WIPI proteins were highly expressed, and AIF1 and PPIF proteins were lowly expressed. The outlined proteins can be further investigated for their involvement in melanoma progression. The expression pattern of the keratinization pathways was different in the comparison of primary and metastasis samples. Furthermore, the upregulation of CD46, MCM9, and TRAF1 proteins and the downregulation of proteins such as PELO and PSMD3 was identified in connection with clinical stages (from AJCC8 stage II. to IV.).

On the basis of the therapy subgroups investigated, a new role of well recognized pathways and proteins in melanoma development was recognized in the therapy response. In our findings, the VEGFA-VEGFR2 pathway (such as NOS3 protein) and RNA splicing mechanism were related to a worse prognosis, and neutrophil degranulation mechanisms and extracellular matrix interactions were connected to better survival during immunotherapy treatments. Furthermore, the upregulation of the MAPK and mTOR signaling components (such as PTEN protein) was associated with short survival, and the significant activation of the RNA mechanisms and stroma signature were connected to better outcome under targeted therapy. These results elucidate new clinical avenues in the underlying molecular mechanisms during the evolution of therapy resistance. Our findings also manifested the essential role regarding the immune system, VEGF inhibitors, spliceosome inhibitors and stroma targets in melanoma treatment. In the coming years, the emphasis will be on discovering and developing predictive biomarkers with hidden and unpredictive melanoma mechanisms during ongoing patient treatments, because this is a strategy to strengthen the immunotherapy response, thereby lowering the impact of developing novel disease mechanisms leading to tumor progression.

Taken together, a broad spectrum of mechanisms involved in tumor progression and therapy response were demonstrated within our study, verifying the complexity of the potentially predictive protein biomarkers in melanoma.

## Figures and Tables

**Figure 1 cancers-13-06105-f001:**
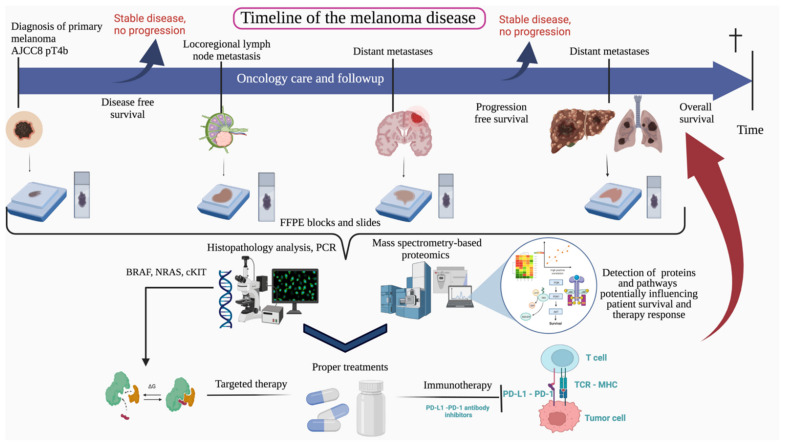
The overview of our study. Scheme illustrating the general working process of our study, showing the timeline of the melanoma disease, from the tumor diagnosis, through progression until the appearance of the distant metastases. This study included 90 FFPE melanoma samples collected during the oncology care and follow-up. The collected samples were submitted for histopathology and protein analysis by high-resolution mass spectrometry. Proteins and pathways were detected indicating survival and therapy response.

**Figure 2 cancers-13-06105-f002:**
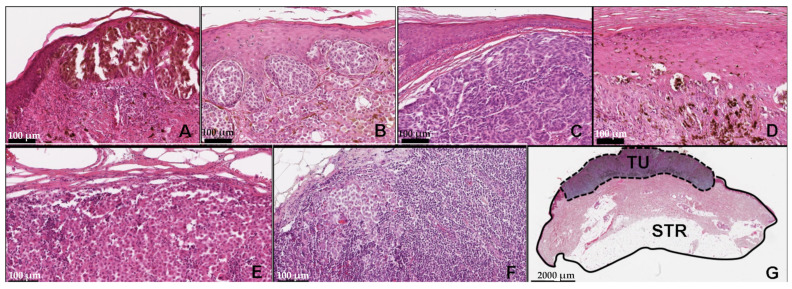
Histopathologic features of the representative cases in the study. Image captures, representing the spectrum of melanoma sample subtypes, including SSM (superficial spreading melanoma) (**A**), SSM with a vertical involvement (superficial spreading melanoma with vertical involvement) (**B**), NM (nodular melanoma) (**C**), ALM (acrolentiginous melanoma) (**D**), as well as both cutaneous (**E**) and lymphatic (**F**) metastases. Each sample also underwent a tumor (TU)/stroma (STR) ratio assessment (**G**) (HE (hematoxylin-eosin); OM (optimal magnitude) 112×).

**Figure 3 cancers-13-06105-f003:**
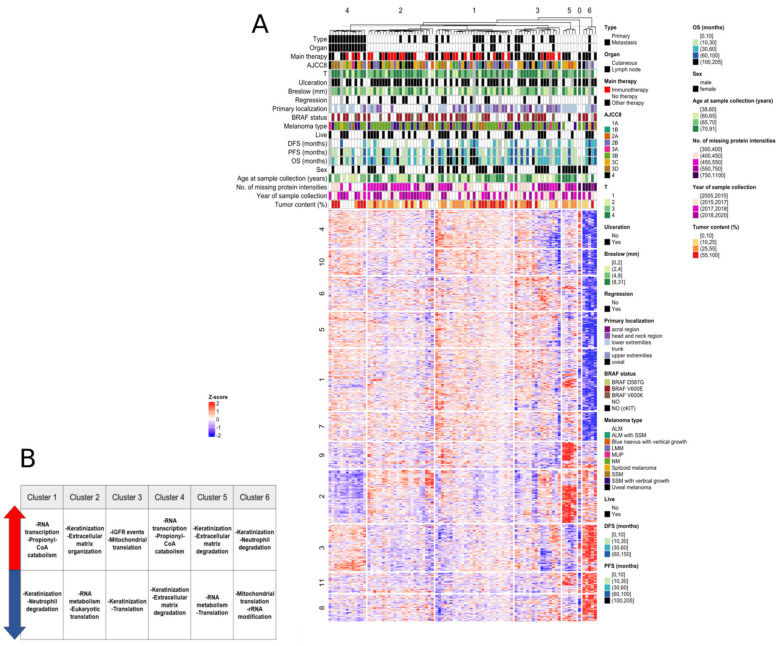
The protein expression patterns in the sample clusters. (**A**) The heatmap of the six detected sample clusters with the clinicopathologic data and the protein expression pattern (top 1000 most variable proteins, ANOVA FDR < 1.05 × 10^−4^). (**B**) The main biological functions linked to the sample clusters. DFS—disease free survival, PFS—progression-free survival and OS—overall survival. Clinical stage based on AJCC8. SSM—superficial spreading melanoma, LMM—lentigo maligna melanoma, NM—nodular melanoma, ALM—acrolentiginous melanoma.

**Figure 4 cancers-13-06105-f004:**
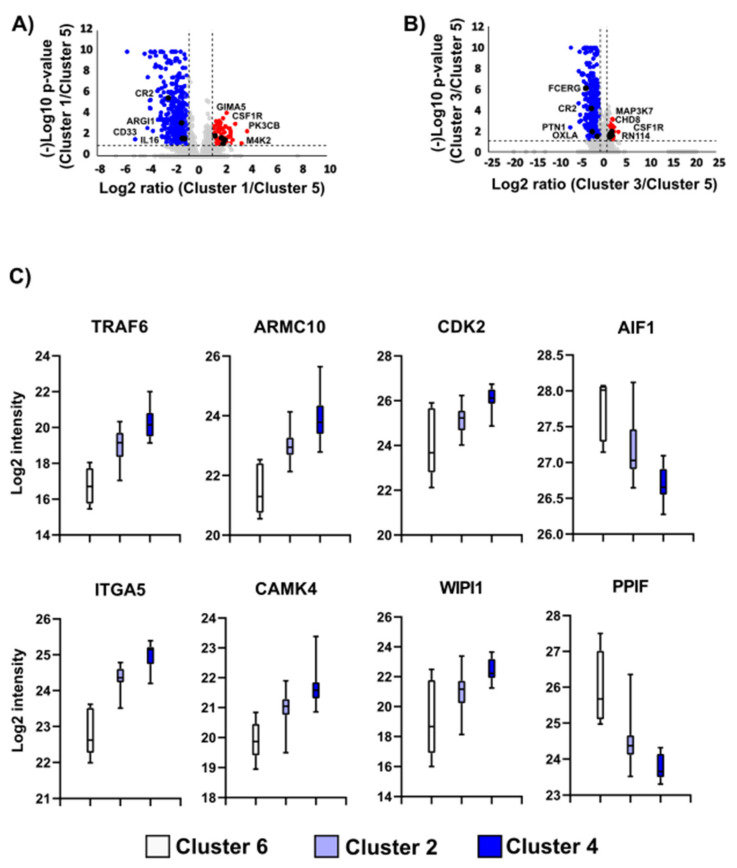
The main up- and downregulated proteins in the sample clusters. (**A**) Volcano plot of sample cluster comparison (between sample cluster 5 and sample cluster 1) with diverse overall survival. (**B**) Volcano plot of sample cluster comparison (between sample cluster 5 and sample cluster 3) with different progression-free survival. (**C**) Examples of the differential expression of the proteins between sample cluster 6, 4 and 2 with different disease-free survival.

**Figure 5 cancers-13-06105-f005:**
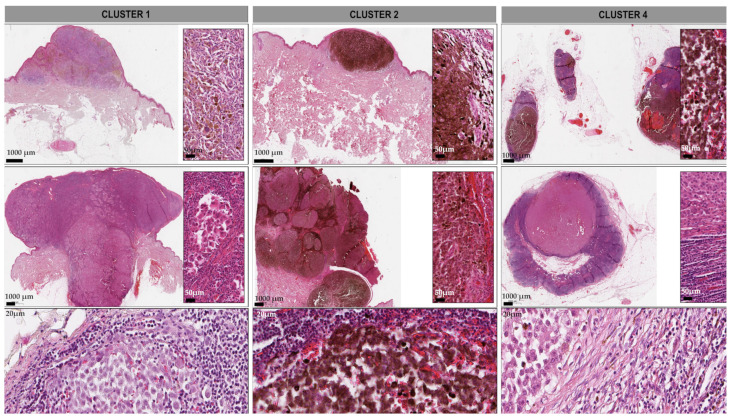
Histopathologic characteristics of the samples in sample cluster 1, 2 and 4. The image represents the spectrum of the morphology of the examined samples, as **Cluster 1** involves vertical growth or de novo nodular melanomas displaying an anaplastic, fusiform or epithelioid phenotype (inserts). Additionally, a few metastases were also included exhibiting the same dedifferentiated morphology. In **Cluster 2**, similar phenotypes were noted, with a pronounced intratumoral melanin pigmentation. **Cluster 4** entailed the majority of the lymph node metastases. OM (optimal magnitude) 112×; HE (hematoxylin-eosin) staining; inserts indicate magnified areas with scale bar 50 μm (Appendix A).

**Figure 6 cancers-13-06105-f006:**
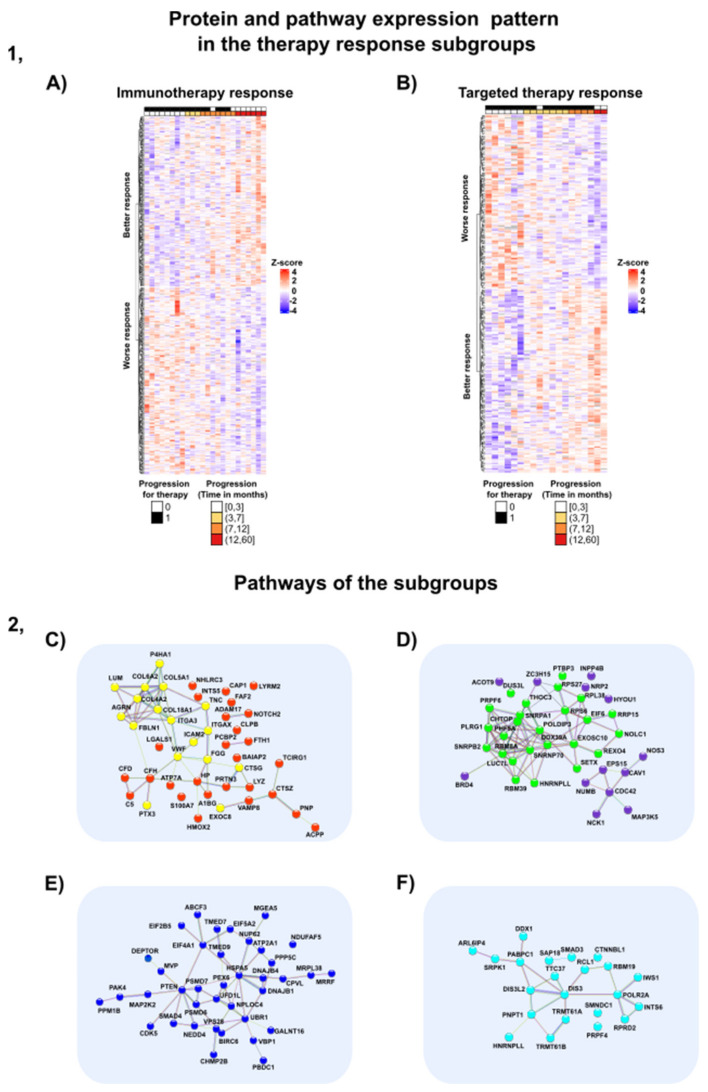
The heatmap of the treatment subgroups and the highlighted pathways. (**1**) Protein and pathway expression pattern in the therapy response subgroups. (**A**) Heatmap of the up- and downregulated proteins in patients with better and worse response in the immunotherapy subgroup. (**B**) Heatmap of the up- and downregulated proteins in patients with better and worse response in the targeted therapy subgroup. (**2**) Protein–protein interaction network of the up- and downregulated proteins in each treatment subgroup. (**C**) Proteins upregulated in patients with better outcome in the immunotherapy subgroup. The yellow color shows the extracellular matrix components, and red indicates the components from the immune system. (**D**) Proteins upregulated in patients with worse response in the immunotherapy subgroup. Proteins from the RNA processing are highlighted in green, whereas the purple color shows the members of the VEGFA-VEGFR2 pathway. (**E**) Proteins upregulated in patients with worse outcomes in the targeted therapy subgroup. The pathway analysis highlighted the key role of proteins involved in metabolic pathways. (**F**) Proteins upregulated in patients with better outcome in the targeted therapy subgroup. Multiple elements of RNA processing can be detected. The list of proteins involved in the aforementioned pathways is given in the Appendix A.

**Table 1 cancers-13-06105-t001:** Clinicopathologic data of the patient cohort.

	Clinicopathological Properties			
Patients	Variable	*n*	Median	Range
Age	Age at primary ^1^	77	64 yrs	54–74 yrs
Survival	DFS (m) ^1^	74	17 months	0–45 months
	PFS (m) ^1^	74	42 months	3–81 months
	OS (m) ^1^	74	51 months	6–96 months
Treatments	**Variable**	** *n* **		**% of total (*n* = 77)**
	Immunotherapy	22		28.57
	CTL4i	2		2.59
	PD-1i, PD-L1i	22		28.57
	Targeted therapy	15		19.48
	BRAFi	15		19.48
	MEKi	14		18.18
	Targeted-plus Immunotherapy	3		3.89
	Other therapies	59		76.62
	irradiation	35		45.45
	chemotherapy	9		11.68
	ECT	4		5.19
	IFN-therapy	13		16.88
	No treatments	18		23.37
Clinical stage (AJCC8)	St. I	7		9.1
	St. II	24		31.16
	St. III	31		40.26
	St. IV	14		18.18
	NA	1		1.3
Tumor samples	**Variable**	** *n* **		**% of total (*n* = 90)**
	Primary tumors	53		58.9
	SSM with vertical growth	15/53		28.3
	SSM	5/53		9.43
	NM	27/53		51
	ALM	3/53		5.66
	ALM with vertical growth	1/53		1.88
	ALM with SSM	1/53		1.88
	LMM	1/53		1.88
	Locoregional lymph. met.	24/90		26.66
	Cutaneous met.	13/90		14.44

The table represents the clinicopathologic parameters of the patients and their selected primary melanomas and metastases included in the pilot patient cohort. DFS—disease-free survival, PFS—progression-free survival, OS—overall survival; CTL4i—cytotoxic T-lymphocyte associated protein inhibitor, PD-1i-programmed cell death protein 1 inhibitor, PD-L1i-programmed death ligand 1 inhibitor, BRAFi—BRAF kinase inhibitor, MEKi—MEK kinase inhibitor, ECT—electrochemotherapy; AJCC8—The 8th edition American Joint Committee on Cancer staging system; yrs—years; SSM—superficial spreading melanoma, LMM—lentigo maligna melanoma, NM—nodular melanoma, ALM—acrolentiginous melanoma, NA—no available data. ^1^ In three cases, the age at primary diagnosis, the disease-free survival, the progression-free survival and the overall survival were not available.

## Data Availability

ANOVA and Tukey’s Honest Significant Difference test results of sample cluster comparisons, as well as Cox regression analysis results for immunotherapy and targeted therapy subgroups, proteomic data summary, *t*-test results (paired/independent) for primary vs. metastasis comparisons as well as ANCOVA results for the comparison of disease stages are provided in Appendix A. The scripts used for the proteomic data normalization, batch effect correction and statistics are available at https://github.com/bszeitz/MM_pilot (accessed on 21 November 2021). The mass spectrometry proteomics data have been deposited to the ProteomeXchange Consortium via the PRIDE [1] partner repository with the dataset identifier PXD028930. Project name is Proteomic Analysis on Paraffine Archived Melanoma. Project accession is PXD028930, project DOI is not applicable, username is reviewer_pxd028930@ebi.ac.uk, and password is TaHGkBGm. Clinical information of individual patients cannot be provided due to the ethics restrictions.

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
