# Peer review of "Deep Proteomic Analysis on Biobanked Paraffine-Archived Melanoma with Prognostic/Predictive Biomarker Read-Out"

_cancers, 2021, doi:10.3390/cancers13236105_

Round 1

Reviewer 1 Report

Szadai et al. conducted a “Deep Proteomic Analysis on Biobanked Paraffin-Archived Melanoma with Prognostic/Predictive Biomarker Read-out” in their current work. The purpose of this study is to identify novel predictive protein biomarkers or protein sets, that may be used to predict therapy response and survival. The authors used mass spectrometry-based proteomics to quantify 7881 protein groups in 90 FFPE samples obtained from 77 individuals (53 primary and 37 metastatic melanoma samples representing both lymphatic and cutaneous metastases). Six significant sample groupings were identified using unsupervised hierarchical cluster analysis. The author did an excellent job of investigating the feasibility of using formalin-fixed paraffin embedded (FFPE) tissue for high-throughput manner.

The author dealt with 90 FFPE samples from 77 patients, which is not a large number, but includes diverse stages of cancer; each stage represents a distinct collection of gene expression, and so there are numerous variables that might affect the outcome of tumour progression and overall survival of the patient. The manner in which the author defines their confounding variable is the major limitation of this study. It is strongly recommended that the authors should clarify this in more detail. Additional suggestions are presented in the section below.

It is not easy to extract and any potential targets from the study. Neither abstract nor conclusion describes the potential name of the target based on the current study results. Most of the sentences are much generalized and discuss only the global finding. It is highly recommended that authors should report the specific names and target protein molecules as a potential candidates.

Line 41-42: It is natural to identify the “distinct protein expression profiles and varying clinicopathologic features”. Please change the statement and give simpler conclusive and concrete findings instead of generalizing sentences.

Line 64-65: Writing personalized medicine too far-fetched with this set of experiments. Please remove the statement.

Line 214-215:  Why the data was acquired at 15000 MS2 resolution.

Line 221-222: What is the fragment mass tolerance in terms of ppm?

What is the reason for keeping met-loss, met-loss+Acetyl variable modification for data search?

Why did the authors use the peptide library search with DDA analysis? What is the rationale for using this analysis, and how does it help to improve the confidence of the data statistically?

It is not clear what is the basis of the selection of Euclidean distance and complete linkage parameter for the hierarchical clustering analysis. Please justify.

Do authors perform any imputation during the proteome discoverer analysis?

The total number of proteins acquired is from a single measure per sample or sample was fractionated to identify the total proteins. What is the number of MS/MS counts identified per sample? Please make the table describing the details of all the values identified. Refer to this article (10.1021/acs.jproteome.0c00670), specifically table 1, to prepare the chart.

Line 364-366: First, authors should report the number of common and unique proteins identified among different clinical stages. Especially the number of unique proteins to the respective condition will be more interesting to report.

The authors choose to use the Euclidean clustering on top of using the soft crusting? Why is it so? As the complete data interpretation depends on clustering, it would be beneficial to report the importance of this selection criteria.

Author Response

Dear Reviewer 1,

We appreciate your comments that help us to further improve our manuscript. Please see the attachment with the cover letter and responses. 

Sincerely, 

Leticia Szadai MD

Reviewer 2 Report

This manuscript used mass spectrometry to discover novel prognostic or treatment predictive protein biomarkers in archival FFPE melanoma samples.

There are several major and minor concerns that I would like to see addressed, which are detailed below.

Major concerns:

1) The manuscript structure, length (28 pages), and figure number (7 Figures) are typical of a thesis, rather than a research article. All sections would greatly benefit from being shortened.

2) The manuscript requires moderate English revisions, particularly the Introduction and Conclusions, where several sentences are incomprehensible. 

3) The indication of which therapeutic agents are being studied is only indicated at the end of the manuscript (section 5. Patients). This should be indicated much earlier.

4) Figure 1, Figure 7 and Table S1 are not mentioned in the text.

5) Graphical abstract: requires revision, as it does not adequately summarise this study. For instance, the mention of diagnostics and stage III disease and the lack of proteomics. The title TOC Figure should also be removed.  

Minor concerns:

INTRODUCTION

1) page 3, lines 86-88: autoimmunity does not occur. At times, immunotherapy can lead to immune-related adverse events that mimic autoimmunity but these are distinct. 

2) page 3, lines 117-121: what are BRAF-subtype melanomas? BRAF mutant? 

3) Figure 1: requires revision. Not all melanomas progress or relapse, and not all patients die. The cellular representation of immunotherapy is inadequate.

RESULTS AND DISCUSSION

4) Table 1: does not follow the journal table format. Treatment specifics need to be included. Abbreviations are not explained. 

5) page 9, lines 340-345: %s are incorrect, and do not add to 100%. 

6) Figure 2:  Needs indication that images are representative cases. Abbreviations are not explained.

7) page 9, section 3.2: which 77 patients were used, and why not all 90?

8) page 9, section 3.2: aren't there 7 clusters in total? The 1st is labeled as cluster 0. 

9) page 15, section 3.7: are tissue samples immediately before treatment? If diagnostic, have there been treatments in between that may alter the proteomic profile of tumors?

CONCLUSIONS

10: This section is confusing, incomprehensible at times, and needs revising. How can the findings in this article be applied to the clinic?

PATIENTS

11) This section is out of place entirely. I recommend moving it to the beginning of the results section instead. 

Author Response

Dear Reviewer 2,

We appreciate your comments that help us to further improve our manuscript. Please see the attachment with the cover letter and responses. 

Sincerely, 

Leticia Szadai MD

Reviewer 3 Report

This is now one of several studies on proteomics of melanoma so is not novel. It would help if new findings could be highlighted and emphasis given to differences in profiles of responders versus non responders . The 8 patients with primary and metastatic melanoma do not appear to have been studied as an entity and could help remove some of the heterogeneity. Discussion on the clinical application would increase interest in the results

Author Response

Dear Reviewer 3,

We appreciate your comments that help us to further improve our manuscript. Please see the attachment with the cover letter and responses. 

Sincerely, 

Leticia Szadai MD

Round 2

Reviewer 1 Report

There are no concerns since the authors have integrated and offered sufficient responses to the questions that were expressed. Following a thorough examination of the whole manuscript, I am satisfied with the current revision.

Reviewer 2 Report

I thank the authors for addressing my concerns and editing the manuscript accordingly. 

I have no further comments.